# The impact of liquefaction disaster on farming systems at agriculture land based on technical and psychosocial perspectives

**Muhammad Basir-Cyio**[1]*, **Mahfudz**[1], **Isrun**[1], **Zeffitni**[2]

**1** Department of Agrotechnology, Faculty of Agriculture, Tadulako University, Palu, Central Sulawesi, Indonesia, **2** Department of Geology, Faculty of Engineering, Tadulako University, Palu, Central Sulawesi, Indonesia

* basircyio@yahoo.com, basircyio@untad.ac.id

**Data Availability Statement:** All relevant data are within the manuscript and its Supporting information files.

## Abstract

This research aims to determine the attitudes of the farmers whose lands are affected by liquefaction in Jono Oge, Central Sulawesi Province, The Republic of Indonesia. The methods used here were integrated survey and experimental design. The survey approach was intended to figure out the attitudes of the farmers viewpoints: (1) to return to their activities on the agricultural lands affected by liquefaction; (2) to consume their own agricultural products; and (3) of their willingness to be relocated. The experimental design approach was used to figure out the effectiveness of organic material input combined with the SP-36 fertilizer. The obtained results were analyzed using the Likert Scale, diversity test, correlational test, and regression test. The results showed that the farmers persevered farming on the lands affected by liquefaction (Index = 88.82%) yet refused to consume their own agricultural products with the reason that corpses remained buried beneath their lands (Index = 27.82%); and they also refused to be relocated (Index = 28.80%). The continued production suitability of the affected land was also investigated. Terrain profile identification results in Jono Oge showed the disaster impact was dominantly landslide as it still showed a clear characteristic horizon between the topsoil and the sub soil. This contrasts to terrain at Petobo, Central Sulawesi Province, where the high mix of the topsoil with the sub soil of agricultural land affected by liquefaction, prevented demarcation of the horizon. The land treatment of organic material and SP-36 fertilizer showed that the combined dose (M) of 40-kg ha$^{-1}$ with P 300-kg ha$^{-1}$ had the highest effect by changing the field pH from 5.7 to 6.41, increased the availability of P and increased the corncob indicator plant weight. Based on these indications, the lands affected by the liquefaction in Jono Oge can still be used as agricultural lands through restoration, from both social and technical aspects.

## Introduction

The earthquake, liquefaction, and tsunami, which simultaneously shook the city of Palu, Donggala Regency, and Sigi Regency, Central Sulawesi Province, The Republic of Indonesia, on September 28, 2018, destroyed infrastructure, and killed no less than 10 thousand people, even more. The—number of lost lives reported by the government is based on recovered

**Funding:** This research was supported by the Board of Professor of Tadulako University.

**Competing interests:** The authors have declared that no competing interests exist.

bodies. The number of victims still buried due to liquefaction and those carried by tsunami waves is difficult to detect because the events are unexpected, and in complete dark conditions. There are an unknown number of victims in addition to former residents of the city of Palu, Sigi and Donggala, as many visitors were at the city of Palu to witness the *Nomoni* traditional event. This local term describes the practise of giving of offerings to the gods around Palu Bay, as a sign of gratitude for the peace of enduring cultural preservation.

Three types of natural disasters that simultaneously occurred, in addition to stopping all economic activity, have also changed the legal aspects of agricultural land ownership affected by the faction. This is due to a shift and displacement of the position of agricultural land between two to three kilometers from the initial position before there was disturbance, including cultivated plants on the land. The shifted geographical locations require new studies and regulations from the government so land ownership and condition can be technically accountable. Earthquakes that occur everywhere will have physical, social, and economic impacts [1, 2]. In fact, traumatic conditions will overshadow people's lives [3], including apathy which threatens to decrease the creativity of the affected people [4].

During the nearly seven months since the natural disaster occurred, there has been no farmer activity on the lands being cleared of disaster debris. There are several factors why farmers are still reluctant to plant crops on land previously used as an agricultural area. There are doubts over the condition of the land under faction which is considered no longer suitable for agriculture. The legal and technical aspects need resolution. There is also the hindering view the damaged land still holds many corpses of victims of liquefaction. Trauma is a physical disorder that takes a long time to heal [5], and on the other hand, farmers are pressured to meet immediate daily needs such as food. Traumatic conditions have a direct impact on the attitude of victims of whatever type of disaster they experienced [6], including decision makers [7]. This condition will have a long-term impact on local socio-economic aspects, especially poverty and declining economic transactions [8, 9]. The agricultural sector must continue to produce both rice and corn agricultural products which society needs for family consumption. In a traumatic transition period, victims need to be encouraged to have a strong vision for their future or a compelling the vision for the future [10].

The principles of sustainable development in the agricultural sector must continue to be considered without reducing concern for safety and environmental sustainability by promoting the value of humanity [11]. The essence of sustainability is the ability to guarantee Humanity by continuing to get what is needed today without compromising the rights of future generations to continue to get their needs met [12]. Ongoing local population security relies on productive agricultural land resources that were affected by liquefaction in Central Sulawesi. The faction of agricultural land which was affected by the liquefaction has experienced a very drastic change, both from environmental and socio-economic aspects. This condition has an impact on farmers' psychocultural needs of spirit, requiring remedies from both scientists and the government so that survivor farmers can continue their key socio-economic roles while maintaining environmental sustainability.

The aim of the scientific study is to provide information from the results of research while at the same time demonstrating impacts of farming practices. Farmers are expected to understand the use of fertilizers and their application techniques on agricultural land affected by liquefaction. In addition, farmers affected by natural disasters can rise again with a new spirit in dealing with environmental conditions that have changed drastically due to natural disasters. Concurrently local governments provide guarantees that local land ownership rights remain a concern of the government that is being resolved through persuasive socialization. It is expected that agricultural production results from convoluted land can be accomplished, both for farmer families' consumption and for sale to traditional markets in order to obtain

education costs for their children. The three main components of sustainability include the community (farmers), economy, and the environment, combined for support by government policies with scientific considerations of research results, are the pillars of strengthening sustainable development [13, 14].

Based on these reasons, a series of studies have been carried out and the results can be disseminated to the farmers and their families to obtain the correct information as well as to commence the revival of their occupational activities.

## Materials and methods

This research was carried out on the land affected by liquefaction caused by the earthquake in Jono Oge, Sigi Regency, by taking a comparison to the liquefaction-affected land area in Petobo, Palu, Central Sulawesi, The Republic of Indonesia. The research began in April to June 2019, when the farmers who were the victims of liquefaction had occupied the temporary housings, built by the government and with assistance by the international NGOs. The area of Petobo agricultural land affected by the earthquake was around 457.10 ha, to the extent the farmers were totally failed in harvesting, while the liquefaction-affected agricultural land in Jono Oge was around 124.20 ha. The map of the research location is presented in Fig 1.

This research assigned human participants to evaluate the effect of earthquake and liquefaction to their income, health. work behavior. They have been well informed that the

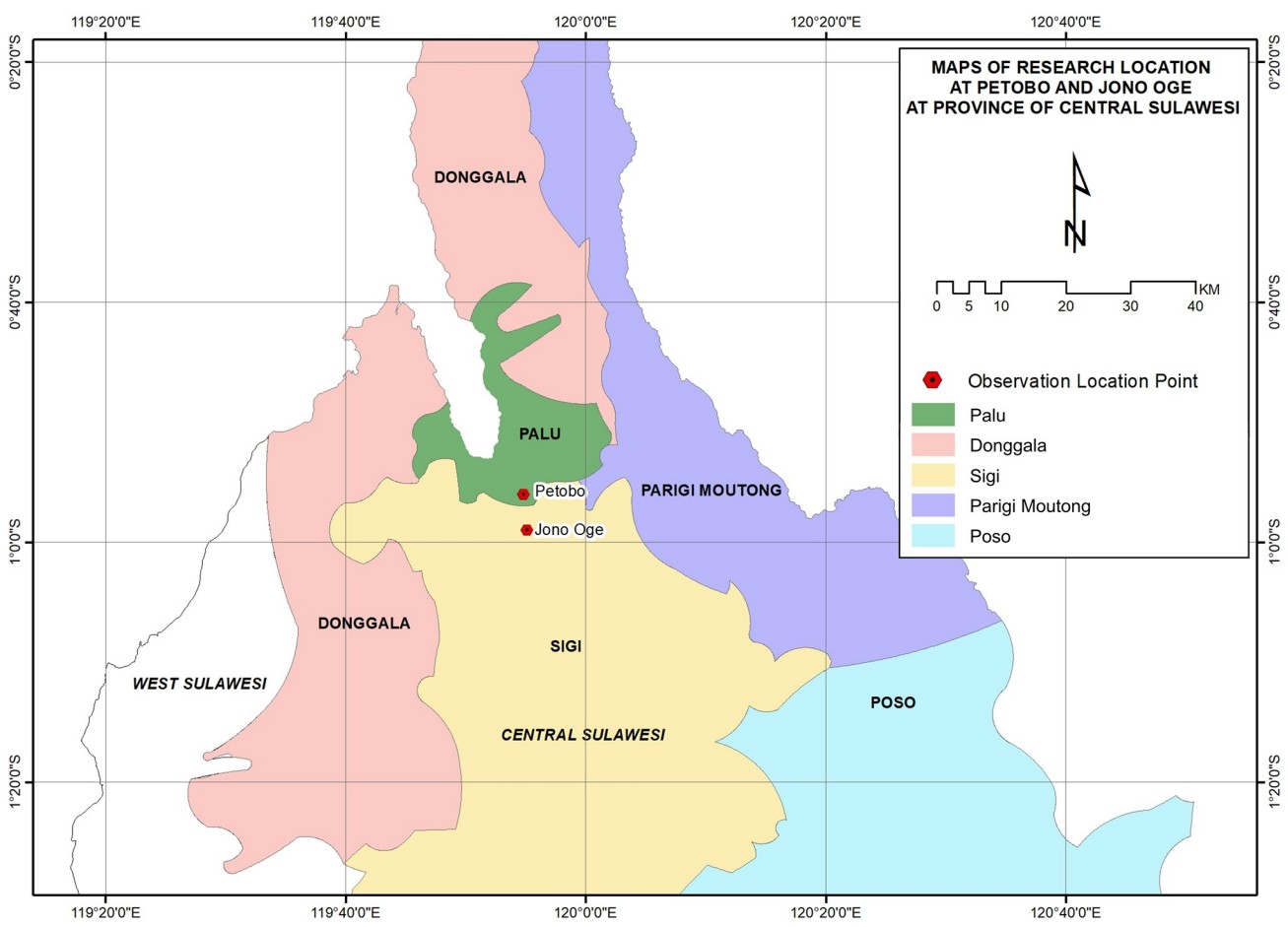

**Fig 1. Locations of research in Jono Oge and Petobo, Central Sulawesi.**

information they provided in the questionnaire will not be affected their activities in The Republic of Indonesia. The study conducted based on UNTAD Research Centre Contract Number: 911.f/UN28.2/PL/2019, and the Ethical Clearance Number 0014/Sinenik/UN.28/ 2019 from the Professor Board of Tadulako University. Consent from the farmers as participants have been obtained, and the written description and personal details of the farmer has been removed and were not published anywhere in the paper.

The Jono Oge agricultural area is the focus of the research, while the Petobo agricultural land is only a comparison, because the two liquefaction patterns between Jono Oge and Petobo are not the same from the perspective of the workable land use for agricultural activities.

## Survey approach

The field survey was carried out in two strategic segments, namely regarding human beings, in this case, the farmers affected by natural disasters, and the condition of the agricultural land environment, where the farmers make a living and financial resources for their families, and the education costs of their children. The determination of the number of the liquefaction-affected respondent farmers used the Slovin Formula, mathematically presented as follows:

$$n = N/\{(1 + N(e)^2\}$$ (1)

**Information**: n = number of Samples

N = Total Population

e = Error Tolerance Limit

Based on this equation, 34 respondents or around 10% of 340 population of farmers were affected by the liquefaction. The respondents were sampled from those who worked in Jono Oge, while those who tried to farm in Petobo were not interviewed and no soil sampling was conducted. They were not interviewed because their conditions were still traumatic, so their answers were often fickle and inconsistent in providing information. Based on these considerations, they were involved in agricultural practices but were not urged to provide an explanation but were directly observed in the field. In addition, farmers affected by liquefaction can rise again with a new spirit in dealing with environmental conditions that have changed drastically due to natural disasters. Farmers are expected to understand the use of fertilizers and their application techniques on agricultural land affected by liquefaction. The two locations show different patterns of land disturbance. Liquefaction in Jono Oge is dominantly "landslide" so that the layer, or horizon, in the soil profile tends not to change to the depth of the solum 0.50 to 2.50 meters. Different results are found in the liquefaction in Petobo, where the pattern and composition of the profile layer undergoes very drastic change, where the topsoil and subsoil of the original soil cannot be identified by its texture, color, or root system.

The in-depth interviews were designed to demonstrate three attitudes of respondent farmers, namely (i) the wish to be back on planting in the liquefaction-affected land; (ii) the willingness to consume agricultural products on the liquefaction-affected land; and (iii) the responses to the government's plan to relocate elsewhere. The farmers' responses to these three attitudes and decisions are identified on Likert Scale [15] with a range of perceptions, "Strongly disagree-1–2–3–4–5 –Strongly agree". By paying attention to the statement items, the respondent farmers affected by liquefaction disaster had to choose numbers 1 to 5 based on the following criteria:

1. strongly agree = 5

2. agree = 4

3. somewhat agree = 3

4. disagree = 2

5. strongly disagree = 1

The determination of the attitudes of the respondent farmers is presented in the Index (%) which is the value of the total score divided by the maximum score times 100 percent [I = (TS/MS) x 100%]. Assessment Intervals are presented in an index range of 0.00%–19.99% (Strongly Disagree); 20.00%–29.99% (Disagree); 40.00%–59.99% (Somewhat Agree); 60.00%–79.99% (Agree) and 80.00%–100.00% (Strongly Agree).

The perceptional data from the respondent farmers affected by natural disasters are strategic facts to be integrated with the data on the assessment of agricultural land resources, a place where farmers obtain a source of life in managing their families, both to meet their daily needs, and related to the sustainability of their children's education. The implementation of the assessment of the agricultural lands affected by the liquefaction disaster, was carried out with three approaches, namely; (i) the soil profile analysis, in order to obtain any information on the topsoil to subsoil conditions, the layers that determine the suitability of the soil to provide the carrying capacity for agricultural activities, according to the type of plant being developed; (ii) the analysis of soil samples in the laboratory, to find out in detail the physical-chemical properties of the soil; and (iii) the direct planting on the liquefaction-affected lands, and by taking soil samples to be assessed in the green house as a comparison of the visualization of the biological condition of the vegetative plant growth with the vegetative growth planted directly on the field.

## Experimental design

The field treatment experiments applied chicken manure fertilizer and SP-36 fertilizer with the aim of improving the physical condition of the liquefaction-affected soil, using the experimental design method. The experimental method [16] determines the effectiveness of chicken manure fertilizer (M) and phosphorus fertilizer (SP-36) which contain $P_2O_5$ (36%). An initial soil sample assessment indicated that the soil was less fertile. The experimental land was set in the form of 1 m x 3 m (3 $m^2$) plots. The chicken manure (M) consisted of four dosage levels, namely (i) $M_0$ (0 t $ha^{-1}$ is equivalent to 0 kg $plot^{-1}$); $M_1$ (10 t $ha^{-1}$ is equivalent to 5 kg $plot^{-1}$), $M_2$ (25 t $ha^{-1}$ is equivalent to 12.5 kg $plot^{-1}$) and $M_3$ (40 t $ha^{-1}$ is equivalent to 20 kg $plot^{-1}$). The test applications of SP-36 synthetic fertilizer consisted of four dosage levels of $S_0$ (0 kg $ha^{-1}$ equivalent to 0 g $plot^{-1}$), $S_1$ (100 kg $ha^{-1}$ equivalent to 50 g $plot^{-1}$), $S_2$ (200 kg $ha^{-1}$ is equivalent to 100 g $plot^{-1}$) and $S_3$ (300 kg $ha^{-1}$ SP-36 is equivalent to 150 g $plot^{-1}$). There were 16 combinations of treatments, namely: $M_0S_0$; $M_0S_1$; $M_0S_2$; $M_0S_3$; $M_1S_0$; $M_1S_1$; $M_1S_2$; $M_1S_3$; $M_2S_0$; $M_2S_1$; $M_2S_2$; $M_2S_3$; $M_3S_0$; $M_3S_1$; $M_3S_2$; and $M_3S_3$. All treatment combinations were repeated three times so that the total experimental units were 48 plots. The observed variables included the soil physical-chemical properties, bulk density, P total, P available, dry weight of plant, phosphorus uptake, and weight of skinless cob.

## Data analysis

**(i) Likert scale data analysis.** The data analyzed from the interviews and questionnaires were tabulated after the respondent farmers selected numbers 1 to 5 on the Likert Scale. The Likert Score Total calculation results for the first statement (i), "The farmers' wish to return to their activities on the liquefaction-affected lands," obtained the total score (TS) = the total number of total subtotals of each answer choice, from a maximum score of 340 and a minimum score of 34. The Maximum Score = 34 x 5 = 170 (number of respondents x highest Likert Score) and The Minimum Score = 34 x 1 = 34 (number of respondents x lowest Likert Score).

Index Calculation (%) = (Total Score/Maximum Score) x 100%. For the statement (i), "The wish to return for planting on the liquefaction-affected land," can be seen in the following results:

The answer of *strongly agree* (SA) = 19 respondents x 5 = 95

The answer of *agree* (A) = 10 respondents x 4 = 40

The answer of *somewhat agree* (SWA) = 4 respondents x 3 = 12

The answer of *disagree* (DA) = 2 respondents x 2 = 4

The answer of *strongly disagree* (SD) = 0 respondent x 1 = 0

Total Score = 151, or Index (151/170) x 100% = 88.8, was between the 00.00%–100% assessment interval which means that the farmers "Very Agree" to return to their activities on their liquefaction-affected lands. The same calculation pattern was also carried out for the other two statements. For the statement (ii), the wish to consume agricultural products of the liquefaction-affected lands obtained TS = 46, or Index (46/170) x 100% = 27.06%, was between the assessment intervals of 20.00%–39.99% which means that the farmers "Disagree" to consume the agricultural products obtained from the liquefaction-affected agricultural lands. Meanwhile, for the statement (iii) about the willingness to be relocated, the TS = 49, or Index (49/170) x 100% = 28.82%, was between the assessment intervals of 20.00%–39.99% which means that the farmers "Do not Agree" to be relocated to other places, or they still wanted to survive on the liquefaction-affected lands to conduct farming.

**(ii) Field assessment, soil sample analysis, and field experiments.** The field assessment was focused on identifying the soil profiles through visual observation of the horizon or the ground layer. In assessing the soil profile, the changes in some physical properties of the soil were carefully examined, especially the changes in the layers due to landslide or liquefaction. There were three soil profiles identified, so that the differences of the layer change pattern in Jono Oge and those around Petobo can be explained. The land in Jono Oge generally experienced landslide or body soil (topsoil—subsoil) position movement to a new coordinate point. In the shifting process, the mix between the topsoil and subsoil layers was minimal, while the agricultural land in Petobo was dominated by the horizon mix process, making observations indicatively difficult to distinguish between the initial topsoil and subsoil because the boundaries between the two layers were visually unidentifiable.

Soil physical-chemical properties were determined through laboratory analysis of composite soil samples on the liquefaction-unaffected land and the liquefaction-affected land, using the Lab Procedures and Methods developed by the University of Wisconsin-Extension-Madison [17].

The observation data of the experimental results were analyzed based on the diversity analysis. When the Fisher-test results showed a measurable treatment effect, then the Duncan's multiple range test followed. The statistical model used was a treatment combination test, and was also presented in a regression analysis [18] using SPSS software and SAS Program version 9.1. The mathematical model of multiple and linear regression was (i) $[Y_{ij} = \mu + \alpha_i + \beta_j + (\alpha\beta)_{ij} + \varepsilon_{ij}]$, and the linear regression analysis was (ii) $[Y = \alpha + bX^2]$

# Result and discussion

## A. Perception

The analyzed data from the results of interviews and questionnaires were tabulated by considering the index value (NI) to simplify determining the conclusions or perceptions of the

**Table 1. Scoring results of the likert value, index, and the attitudes of the respondent farmers.**

| No. | Perceptional Scope | Score | | Total Score | Indexed (%) | Assessment Interval (%) | Farmers' Attitude |
|-----|-------------------|-------|-----|-------------|-------------|------------------------|-------------------|
| | | Min | Max | | | | |
| 1 | Wish to Return to the Activities on the Lands Affected by Liquefaction | 34 | 170 | 151 | 88.80 | 0–19.99 (SDA) | Strongly Agree (SA) |
| | | | | | | 20–39.99 (DA) | |
| | | | | | | 40–59.99 (SWA) | |
| | | | | | | 60–79.99 (A) | |
| | | | | | | 80–100 (SA) | |
| 2 | Consuming the Agricultural Products from the Lands Affected by Liquidation | 34 | 170 | 46 | 27.06 | 0–19.99 (SDA) | Disagree (DA) |
| | | | | | | 20–39.99 (DA) | |
| | | | | | | 40–59.99 (SWA) | |
| | | | | | | 60–79.99 (A) | |
| | | | | | | 80–100 (SA) | |
| 3 | Resettlement to the locations which are not affected by natural disasters | 34 | 170 | 49 | 28.82 | 0–19.99 (SDA) | Disagree (DA) |
| | | | | | | 20–39.99 (DA) | |
| | | | | | | 40–59.99 (SWA) | |
| | | | | | | 60–79.99 (A) | |
| | | | | | | 80–100 (SA) | |

respondent farmers whose agricultural lands were affected by liquefaction. By paying attention to the statement items, the respondents had to choose numbers 1 to 5. The total Likert Score can be seen from the calculation of each part which was assessed, as presented in Table 1.

From the tabulation of Likert Scale and Index by observing the intervals of the respondents' attitude assessment of the three attitude statements, it is revealed that the farmers who have been trying to farm on the liquefaction-affected lands strongly agree to continue to manage their own agricultural lands, even though it has been affected by liquefaction (88.82%). The farmers still hope to be able to return to work their liquefaction-affected agricultural lands. This attitude shows that sociocultural has the power to influence one's personal decisions [19, 20], including the farmers who are affected by the liquefaction natural disasters in Jono Oge. Meanwhile, the statements regarding the willingness to consume the agricultural products obtained from the liquefaction-affected agricultural lands, they said they did not agree, yet wanted to sell to the markets (Index of 27.08%). The reluctance to consume agricultural products is based on the consideration of the need for money so that it tends to be sold or because there is a wrong perception [21, 22]. The reluctance to consume agricultural products on the liquefaction-affected lands is caused by the misperception human corpses still buried beneath the ground have the potential to be absorbed by plant roots. This attitude reflects the low-level knowledge of the farmers. A person's education can affect the mindset and also the rationality in seeing a problem [23, 24]. The third attitude is to refuse the resettlement to another safer location. This is shown by the obtained Index at 28.82% (Disagree) to leave the location adjacent to the liquefaction-affected lands. The cultural power is very decisive in social cohesion (socio-cohesion) which applies in a particular community [25, 26].

## B. Observation data profile and soil chemical-physical properties

The liquefaction of agricultural lands in Jono Oge and Petobo (Fig 2) has completely stopped the agricultural activities which have affected the plight of the farmers. For the liquefaction victims who have survived natural disasters, it has been difficult to obtain a livelihood because they depend only on agricultural production.

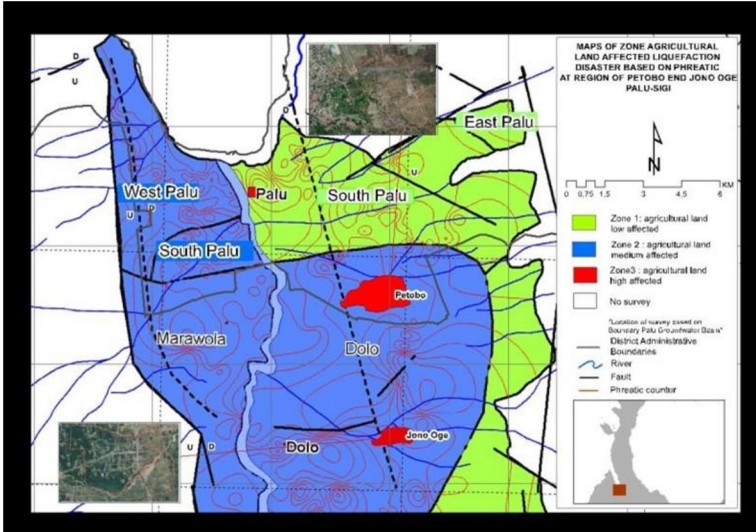

**Fig 2. The liquefaction-affected agricultural areas of Jono Oge and Petobo, Central Sulawesi.**

The current condition of the liquefaction-affected agricultural lands is still neglected, and there has yet been no substantial agricultural activity. The farmers doubt starting to work on their agricultural lands due to a sense of trauma, furthermore, the resolve of the government is also needed. The farmers also need scientific information from researchers, both in terms of soil fertility and groundwater conditions [27, 28], to ensure that the liquefaction-affected lands can still be used for planting. The profile observation results on the liquefaction-affected land in Jono Oge show that the soil has a horizon position which is still relatively the same between the liquefaction-unaffected Entisol layers. The liquefaction-affected soil in Jodo Oge (Fig 3B) still has features which are not different from the landslide-unaffected soil (Fig 3A), except for the only slightly gray color in the landslide lands. Some landslide processes are randomized, yet some move to a certain depth without mixing [29, 30]. The analytical results of the subsoil physical properties in the two profiles do not differ, especially in the texture composition

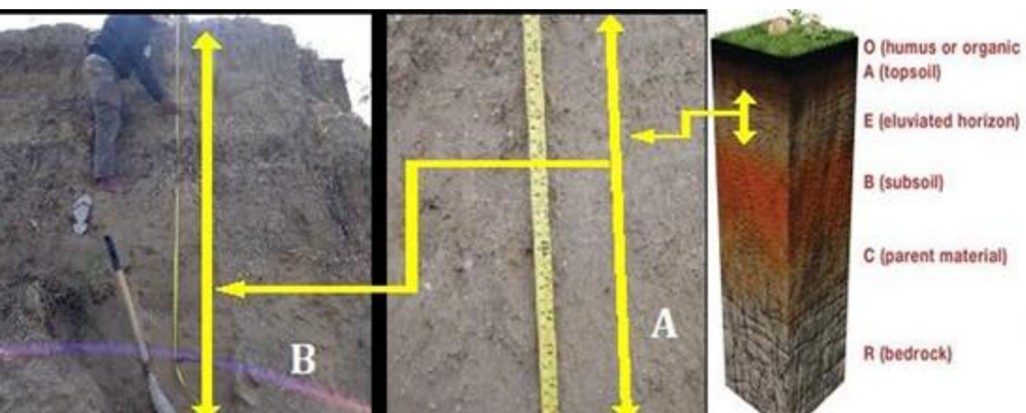

**Fig 3. The Profile of the liquefaction-affected soil (Landslide-B), and the Profile of the liquefaction-unaffected soil (Landslide-A).**

**Table 2. Analysis of the physical-chemical properties of the initial soil samples.**

| No. | Parameter | Unit | Score | Information |
|---|---|---|---|---|
| 1 | Sand | % | 60.9 | |
| 2 | Loam | % | 10.2 | Loamy clay sand |
| 3 | Clay | % | 28.9 | |
| 5 | Bulk density | g/cm$^3$ | 1.6 | |
| 7 | C-organic | % | 1.02 | Low |
| 8 | N-Total | % | 0.16 | Low |
| 9 | C/N | | 8.5 | Low |
| 10 | pH $H_2O$ (1:2.5) | | 5.8 | Somewhat Acid |
| 11 | pH KCl (1:2.5) | | 4.9 | |
| 12 | $P_2O_5$ (HCl 25%) | mg/100g | 24.08 | Low |
| 13 | $P_2O_5$ (Bray I) | Ppm | 10.33 | Low |
| 14 | $K_2O$ (HCl 25%) | me/100g | 29.41 | Low |
| 15 | $Ca^{2+}$ | me/100g | 4.65 | Low |
| 16 | $Mg^{2+}$ | me/100g | 0.43 | Low |
| 17 | $K^+$ | me/100g | 0.25 | Low |
| 18 | $Na^+$ | me/100g | 0.17 | Low |
| 19 | CEC | me/100g | 23.27 | Low |
| 20 | BS | % | 23.63 | Low |
| 21 | Al-dd | me/100g | 0.35 | |
| 22 | H-dd | me/100g | 0.3 | |

Information: Analyzed at the Agrotechnology Laboratory of the Faculty of Agriculture, Tadulako University (2019)

(Table 2). The soil texture can be used as an indicator in assessing the soil physical properties [31, 32]. The comparison results of the soil profiles have found no significant distinguishing characteristics, ranging from topsoil to the 2.5-m solum. The color changes in the landslide-affected lands indicate an infiltration of organic compounds from the soil surface (layer 0) into the topsoil, during the landslide event. The infiltration that carries the organic compounds affects the soil color changes in the subsoil, especially in the illuviation layer [33].

The similarity of the soil physical properties, especially the texture and position of the topsoil and subsoil in the liquefaction-affected soil and the unaffected one, shows strong cementation between the particles in topsoil and subsoil. Cementation is a determining factor in the strength of interparticle adhesion force work [34, 35]. The movements and shifts that occur in the uncemented layer underneath 2.5 meters, where the level of water saturation is maximum, do not affect the topsoil layer on the land that can be used for planting. Thus, technically it is still possible to be processed to produce agricultural products. This is also strengthened by the assessment results of the soil chemical properties in the laboratory which show no significant difference between the landslide-affected soil and the landslide-unaffected soil (Table 2).

In Table 2, it can be seen that the soil texture class is in the category of "loamy clay sand" with the composition of 60.90% sand, 10.20% dust, and 28.90% clay. The bulk density (ρ) of the soil is 1.60 g cm$^{-3}$. The sandy texture class has high porosity so that the water holding capacity is low [36, 37]. The analysis results of several chemical properties, namely the pH ($H_2O$) is 5.33, and the pH (KCl) is 4.66, the $Al_{ex}$ content is 0.35 me/100 g, with the low exchangeable base ion which are $Ca^{2+}$ (4.65 me)/100 g), $Mg^{2+}$ (0.43 me/100 g), $K^+$ (0.25 me/100 g), and $Na^+$ (0.17 me/100 g). The N-total value is low (0.16%), as well as the C-organic (1.02%). The C/N ratio is low, the P-available is low (10.30 ppm), the P-total is moderate

(24.08 mg / 100 g), the K-total is moderate (29.41 mg/100 g), and the Cation Exchange Capacity (CEC) (23.27 me/100 g) is low. Based on those soil properties, the liquefaction-affected lands have low fertility, both before and after the liquefaction disaster. This condition shows that the liquefaction-affected lands from the physical aspect can still be used as agricultural areas, however it requires input to increase the nutrient availability in order to optimize the soil health and the carrying capacity for the plant growth. The administration of the organic materials input can improve the soil structure and the water holding capacity [38, 39].

Providing input from organic material (manure) and SP-36 Fertilizer also improves the propagation chemical properties of the soil, especially pH, available P, dry weight of plant tissue, uptake of P and also the weight of corn cobs. The experimental results showed that all doses of organic fertilizer and SP-36 and their combination had a significant effect on the observed variables (Table 3). Organic matter, in addition to being a source of OH-1 groups, can also release other organic compounds [40].

The combination of the organic materials and the SP-36 significantly affects the physical-chemical properties of the liquefaction-affected soil. The used dosage is in accordance with the organic materials availability and the farmers availability who are dispersed by natural disasters in purchasing the SP-36 fertilizers which are easily obtained in the market. The treatments required for the organic fertilizer and the SP-36 are presented in Fig 4. The autocorrelation between the dependent variables (Table 4) shows the nutrients released from the organic fertilizer and the SP-36 into the rhizosphere adopted by the plant roots into the tissue, influence each other in the plant growth process. The interaction between the organic materials and the synthetic fertilizer accelerates the nutrients availability required by the growth of vegetative and generative plants [41, 42].

The administration of organic fertilizer and SP-36 fertilizer can significantly increase the soil pH ($H_2O$). The increased soil pH ($H_2O$), due to chicken manure and SP-36 fertilizer, is optimum at the combination of 40-t ha$^{-1}$ ($P_3$) dosage of organic fertilizer, and 300-kg ha$^{-1}$ ($P_3S_3$) dosage of SP-36 fertilizer. Before being treated, the soil pH ($H_2O$) of 5.35 increased to 6.38 or increased by 1.04 points after treatment. The changes in pH from slightly acidic to neutral condition affect the macro nutrients availability, such as P, K, and Ca, and suppress their solubility, such as $Al^{3+}$, $Fe^{3+}$, and $Mn^{2+}$. The dynamics of the soil chemical properties occur due to the release of hydroxyl ions ($OH^{-1}$) from the decomposition of the organic materials, as a counterweight to the dynamics of $H^+$ ions in the soil solution [43, 44]. The presence of $OH^-$ ions will stabilize the activity of $H^+$ ions so that the acidity of the soil is controlled under conditions which are somewhat neutral to neutral [45]. Under the neutral pH conditions, the micro nutrient ions which have the potential to fix the macro nutrient ions are also controlled [46], thus the nutrients availability becomes high so that it is easily absorbed by the plant roots [47]. The adequacy of macro and micro nutrients needed by plants depends on their availability in the soil solutions around the rhizosphere [48]. The higher the availability, the more potential

**Table 3. Results of analysis of variance effect (F) of organic fertilizers and SP-36 fertilizers on variable dependents.**

| Sources of Variants | F-calculated | | | | | |
|---|---|---|---|---|---|---|
| | pH-H$_2$O | P-total (mg/100-g soil | P-available (ppm) | Dry weight plant tissue (g plot$^{-1}$) | P-uptake (mg Kg$^{-1}$) | Corncob Weight (g plot$^{-1}$) |
| Group | 0.490306 | 0.253968 | 2.493292 | 0.667025 | 0.29509 | 0.58458 |
| Interaction (MxP) | 1327.326** | 1498346 ** | 634022.2 ** | 2931.163 ** | 143.14 ** | 54.8875 ** |
| Manure (M) | 4688.13** | 4762049** | 2237699** | 9064.184 ** | 447.63 ** | 108.147 ** |
| Fertilizer SP-36 (P) | 1898.6** | 2301411 ** | 820924.2** | 4948.169 ** | 259.009 ** | 157.269 ** |

** Treatment is significant at the 0.01 level (2-tailed).

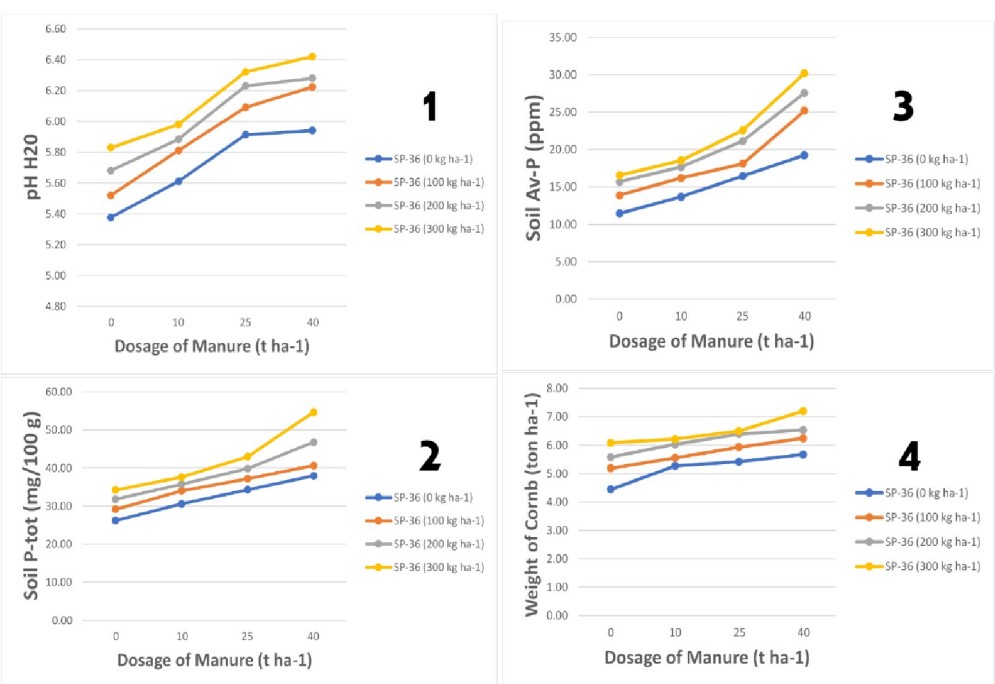

**Fig 4. The impact of dosage interaction between the organic materials with the SP-36 fertilizer on (1) pH (H$_2$O), (2) P-total, (3) P-available, and (4) the weight of corncobs.**

the root absorption process will also occur [49]. The accumulation of nutrients in the ribosome simplifies the distribution from xylem to phloem after undergoing a process of synthesis in the plant leaf tissue cells [50]. This condition certainly still occurs in the liquefaction-affected agricultural lands so that the corn plants, as the indicator plants, grow normally due to the carrying

**Table 4. Correlation analysis results of pH-H$_2$O, P-total, P-available, P absorption, and the weight of cobs.**

| Variable Indicators | | pH-H$_2$O | P-total | P-available | P-absorption | Corncob Weight |
|---|---|---|---|---|---|---|
| pH-H$_2$O | Pearson Correlation | 1 | 0.922(**) | 0.930(**) | 0.960(**) | 0.923(**) |
| | Sig. (2-tailed) | | 0.000 | 0.000 | 0.000 | 0.000 |
| | N | 16 | 16 | 16 | 16 | 16 |
| P- total | Pearson Correlation | 0.922(**) | 1 | 0.972(**) | 0.926(**) | 0.897(**) |
| | Sig. (2-tailed) | 0.000 | | 0.000 | 0.000 | 0.000 |
| | N | 16 | 16 | 16 | 16 | 16 |
| P-available | Pearson Correlation | 0.930(**) | 0.972(**) | 1 | 0.942(**) | 0.926(**) |
| | Sig. (2-tailed) | 0.000 | 0.000 | | 0.000 | 0.000 |
| | N | 16 | 16 | 16 | 16 | 16 |
| P-uptake | Pearson Correlation | 0.960(**) | 0.926(**) | 0.942(**) | 1 | 0.916(**) |
| | Sig. (2-tailed) | 0.000 | 0.000 | 0.000 | | 0.000 |
| | N | 16 | 16 | 16 | 16 | 16 |
| Corncob Weight | Pearson Correlation | 0.923(**) | 0.897(**) | 0.926(**) | 0.916(**) | 1 |
| | Sig. (2-tailed) | 0,000 | 0,000 | 0,000 | 0.000 | |
| | N | 16 | 16 | 16 | 16 | 16 |

** Correlation is significant at the 0.01 level (2-tailed).

capacity of the soil physical-chemical properties after receiving input from the organic fertilizer and SP-36 fertilizer as shown in Fig 4.

The administration of 40-t ha$^{-1}$ and SP-36 300-kg ha$^{-1}$ (P$_3$S$_3$) dosage of the organic fertilizer had the highest effect on increasing the soil pH, P-total, P-available, and the weight of corncobs. The significant response of plants indicated that the liquefaction-affected lands required input in increasing the nutrients availability needed by plants [51, 52]. The response difference between the treatment combination of P$_0$S$_0$ and P$_3$S$_3$ seemed significant between both the blue line and the yellow line, for the pH (1), P-total (2), P-available (3), and the weight of corncob (4). Increasing the soil pH to the neutral condition, by administering organic materials, stabilized the dynamics of the chemical properties of the soil solution, in which the micro nutrients solubility was minimized and also reduced, thus directly increased the phosphorus nutrients availability [53, 54].

## Conclusion

The farmers in Jono Oge continued farming in the liquefaction-affected areas, however they were reluctant to consume their own agricultural products because they thought there were still human corpse victims of liquefaction and earthquake, beneath the surface. The liquefaction-affected dual farming areas were different in the impact patterns, with the Jono Oge dominantly being landslide, so that the identifiers between the topsoil and the sub soil remained easily identified; while there had been a very high mixture on the agricultural land in Petobo, thus the horizon of the soil's profile was difficult to identify. The physical-chemical condition of the liquefaction-affected soil in Jono Oge was able to be restored by administering organic materials and the SP-36 fertilizer, however, the farmers needed the support from the three parties, namely the government related to regulations, scientists to provide dissemination of the research results, and NGO activists to provide advice or assistance including to revitalize their spirit to change for a better life in the future.

## Supporting information

**S1 Data.**
(ZIP)

**S1 File.**
(ZIP)

## Acknowledgments

This research was conducted by the collaboration with Professor Timothy Roberts from the Hunter Innovation and Science Hub of the University of Newcastle, Australia. We also wish to acknowledge Melanie Ball for English editing of the manuscript.

## Author Contributions

**Conceptualization:** Muhammad Basir-Cyio, Mahfudz, Zeffitni.

**Data curation:** Muhammad Basir-Cyio.

**Formal analysis:** Muhammad Basir-Cyio, Isrun.

**Investigation:** Muhammad Basir-Cyio.

**Methodology:** Muhammad Basir-Cyio.

**Project administration:** Isrun, Zeffitni.

**Resources:** Mahfudz.

**Software:** Zeffitni.

**Supervision:** Isrun, Zeffitni.

**Validation:** Mahfudz.

**Writing – original draft:** Muhammad Basir-Cyio.

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
