## [Decision Letter · Decision Letter 0]

26 May 2020

PONE-D-20-00593

The Impact of Liquefaction Disaster on Farming Systems at Agriculture Land Based on Technical and Psychosocial Perspectives

PLOS ONE

Dear Dr. Muhammed Basir-Cyio

Thank you for submitting your manuscript to PLOS ONE. After careful consideration, we feel that it has merit but does not fully meet PLOS ONE’s publication criteria as it currently stands. Therefore, we invite you to submit a revised version of the manuscript that addresses the points raised during the review process.

Many thanks for submitting your manuscript to PLOS One

The manuscript was reviewed by experts in the field and they have suggested that some modifications be made prior to acceptance.

If you could write a response to reviewers that will aid to expedite the review when resubmitted

I wish you all the best with your revisions

Hope you are keeping safe and well in these difficult times

We look forward to receiving your revised manuscript.

Kind regards,

Simon Clegg, PhD

Academic Editor

PLOS ONE

3. We note that Figures 1 and 2 in your submission contain map images which may be copyrighted.

a. You may seek permission from the original copyright holder of Figures 1 and 2 to publish the content specifically under the CC BY 4.0 license. 

Reviewers' comments:

Reviewer's Responses to Questions

**Comments to the Author**

1. Is the manuscript technically sound, and do the data support the conclusions?

Reviewer #2: Partly

Reviewer #3: Partly

2. Has the statistical analysis been performed appropriately and rigorously? 

Reviewer #2: No

Reviewer #3: Yes

3. Have the authors made all data underlying the findings in their manuscript fully available?

Reviewer #2: Yes

Reviewer #3: No

4. Is the manuscript presented in an intelligible fashion and written in standard English?

Reviewer #2: Yes

Reviewer #3: Yes

5. Review Comments to the Author

Reviewer #2: The objective of the study is very broad not clearly elaborated. It doesnot specify what the study is going to investigate.

The methodology is weak. The perception survey is carried out only in one site. Two sites would be better for comparison as well as for better concrete results. The experimental methods does not indicate where and why the experiments were conducted.

Some references is missing. For instance, Joshi et al 2015 is missing.

There are many typos in the paper

Implications of research findings and limitation of the studies are not clearly outlined.

Reviewer #3: This paper is quite interesting. It takes a multi-disciplinary approach towards understanding how farmer attitudes towards continued cultivation of liquefaction-affected rural fields are shaped by likely presence of the dead, embedded in areas of landsliding and liquefaction, more than the actual soil physical-chemical properties of the soil. IN a general sense, the farmers are willing to continue to farm in these areas, and unwilling to move elsewhere, but unwilling to consume their own products grown in these fields. The authors attribute the latter to suggest that farmers are worried about 'root uptake' of the deceased into plants, and go on to make the unfortunate mistake of stating that this reflects the "low-level knowledge" of the farmers. The latter statement is arrogant and should be removed; there are many types of knowledge, including experience, and it is not impossible that degrading corpses also contribute locally to changes in soil chemistry. I understand why farmers might not want to consume crops grown amongst the dead for a variety of reasons, and cannot say I would be different!

That said, I think the paper is really interesting and I want to encourage the authors. I have made a series of notes on the PDF that I hope will help them revise this work. I don't know why the transcripts can not be made available with all identifying information removed, because these transcripts are the actual data to support what the authors are proposing here.

6. PLOS authors have the option to publish the peer review history of their article (what does this mean?). If published, this will include your full peer review and any attached files.

Reviewer #2: No

---

## [Author Response · Author response to Decision Letter 0]

30 Sep 2020

Reviewers' comments:

Reviewer's Responses to Questions

Comments to the Author

1. Is the manuscript technically sound, and do the data support the conclusions?

Reviewer #2: Partly

Reviewer #3: Partly

We have reviewed and corrected the suggestions and corrections from the Review Team. Thank you for all the help and advice.

2. Has the statistical analysis been performed appropriately and rigorously?

Reviewer #2: No

Reviewer #3: Yes

We have read and reviewed the Statistical Analysis in the article. From the results of the re-examination, it was found that the Statistical Analysis that we wrote was appropriate. Re-examination has been done to the maximum.

3. Have the authors made all data underlying the findings in their manuscript fully available?

Reviewer #2: Yes

Reviewer #3: No

Thank you for the advice from the Reviewer Team. In our opinion, the data underlying the findings are contained and written in the manuscript as a whole, without exception.________________________________________

4. Is the manuscript presented in an intelligible fashion and written in standard English?

Reviewer #2: Yes

Reviewer #3: Yes

The language used in the manuscript is standard English which is easy to understand and unambiguous. Thank you for the review.________________________________________

5. Review Comments to the Author

Reviewer #2: The objective of the study is very broad not clearly elaborated. It doesnot specify what the study is going to investigate.

The methodology is weak. The perception survey is carried out only in one site. Two sites would be better for comparison as well as for better concrete results. The experimental methods does not indicate where and why the experiments were conducted.

Some references is missing. For instance, Joshi et al 2015 is missing.

There are many typos in the paper

Implications of research findings and limitation of the studies are not clearly outlined.

Reviewer #3: This paper is quite interesting. It takes a multi-disciplinary approach towards understanding how farmer attitudes towards continued cultivation of liquefaction-affected rural fields are shaped by likely presence of the dead, embedded in areas of landsliding and liquefaction, more than the actual soil physical-chemical properties of the soil. IN a general sense, the farmers are willing to continue to farm in these areas, and unwilling to move elsewhere, but unwilling to consume their own products grown in these fields. The authors attribute the latter to suggest that farmers are worried about 'root uptake' of the deceased into plants, and go on to make the unfortunate mistake of stating that this reflects the "low-level knowledge" of the farmers. The latter statement is arrogant and should be removed; there are many types of knowledge, including experience, and it is not impossible that degrading corpses also contribute locally to changes in soil chemistry. I understand why farmers might not want to consume crops grown amongst the dead for a variety of reasons, and cannot say I would be different!

That said, I think the paper is really interesting and I want to encourage the authors. I have made a series of notes on the PDF that I hope will help them revise this work. I don't know why the transcripts can not be made available with all identifying information removed, because these transcripts are the actual data to support what the authors are proposing here.

We have written down the suggestions and corrections from the Review Team in the manuscript. We are very grateful for the help, direction, and correction from the Review Team in helping to refine the manuscript which aims to perfect the manuscript we sent.________________________________________

6. PLOS authors have the option to publish the peer review history of their article (what does this mean?). If published, this will include your full peer review and any attached files.

Do you want your identity to be public for this peer review? For information about this choice, including consent withdrawal, please see our Privacy Policy.

Reviewer #2: No

---

## [Decision Letter · Decision Letter 1]

25 Nov 2020

PONE-D-20-00593R1

The Impact of Liquefaction Disaster on Farming Systems at Agriculture Land Based on Technical and Psychosocial Perspectives

PLOS ONE

Dear Dr. Basir-Cyio

Thank you for submitting your manuscript to PLOS ONE. After careful consideration, we feel that it has merit but does not fully meet PLOS ONE’s publication criteria as it currently stands. Therefore, we invite you to submit a revised version of the manuscript that addresses the points raised during the review process.

Many thanks for resubmitting your manuscript to PLOS One

Although one reviewer is happy, one is unhappy with the way in which you have handled their comments

Can you please check over the comments of the reviewer from the previous submission, and check that you include each of their comments

If you could write a response to reviewers, that will expedite the review when the manuscript is resubmitted

I wish you the best of luck with your revisions

Thanks

Simon

We look forward to receiving your revised manuscript.

Kind regards,

Simon Clegg, PhD

Academic Editor

PLOS ONE

Reviewers' comments:

Reviewer's Responses to Questions

**Comments to the Author**

1. If the authors have adequately addressed your comments raised in a previous round of review and you feel that this manuscript is now acceptable for publication, you may indicate that here to bypass the “Comments to the Author” section, enter your conflict of interest statement in the “Confidential to Editor” section, and submit your "Accept" recommendation.

Reviewer #2: (No Response)

Reviewer #3: (No Response)

2. Is the manuscript technically sound, and do the data support the conclusions?

Reviewer #2: Yes

Reviewer #3: Partly

3. Has the statistical analysis been performed appropriately and rigorously? 

Reviewer #2: Yes

Reviewer #3: Yes

4. Have the authors made all data underlying the findings in their manuscript fully available?

Reviewer #2: Yes

Reviewer #3: Yes

5. Is the manuscript presented in an intelligible fashion and written in standard English?

Reviewer #2: Yes

Reviewer #3: No

6. Review Comments to the Author

Reviewer #2: I suggest to edit few minor language errors

Reviewer #3: Sadly, I cannot recommend publication. The paper is plagued with grammatical errors, beginning with the title, and persisting throughout. The research method failed to provide a relevant pre-earthquake baseline with which to evaluate the results - for example, what % of farmers consumed the agricultural products from their lands prior to the earthquakes and thus how have the earthquakes changed this? The authors still adhere to the statement that "This attitude reflects the low-level knowledge of the farmers" and thus the authors' perspective of what constitutes 'knowledge' remain unchanged despite my prior recommendations. The paper floats between an agricultural piece on fertilizer and a social science piece on cultural attitudes but ultimately neither section fulfills its potential nor are clear lines drawn to connect these attributes. The authors should consider submitting this work to a lower tier journal. I really wanted to be able to support publication of this work because it has some interesting components, but ultimately I cannot.

7. PLOS authors have the option to publish the peer review history of their article (what does this mean?). If published, this will include your full peer review and any attached files.

Reviewer #2: No

Reviewer #3: No

---

## [Author Response · Author response to Decision Letter 1]

21 Dec 2020

6. Review Comments to the Author

Reviewer #2: I suggest to edit few minor language errors

Reesponse to reviewer #2:

Author : Thank you for reviewer 2. We have corrected some minor errors in the manuscript by re-reading it carefully and details according this suggestion. 

Reviewer #3: Sadly, I cannot recommend publication. The paper is plagued with grammatical errors, beginning with the title, and persisting throughout. The research method failed to provide a relevant pre-earthquake baseline with which to evaluate the results - for example, what % of farmers consumed the agricultural products from their lands prior to the earthquakes and thus how have the earthquakes changed this? The authors still adhere to the statement that "This attitude reflects the low-level knowledge of the farmers" and thus the authors' perspective of what constitutes 'knowledge' remain unchanged despite my prior recommendations. The paper floats between an agricultural piece on fertilizer and a social science piece on cultural attitudes but ultimately neither section fulfills its potential nor are clear lines drawn to connect these attributes. The authors should consider submitting this work to a lower tier journal. I really wanted to be able to support publication of this work because it has some interesting components, but ultimately I cannot.

Response to reviewer #3:

Author: (A) In principle, the method we use has provided steps that can answer problems found in the field, especially related to the comparison or comparison of how many farmers consume their agricultural products before and after the earthquake and liquefaction. Before the disaster 100% of the farmers consumed their agricultural products and after the disaster no one wanted to consume their agricultural products because they believed that the land where they were farming there were still many corpses. Therefore they assumed that the remains of the corpses had been absorbed by plants. So that they are considered to have polluted the land and agricultural products, which is why they are reluctant to consume their agricultural products, especially to provide family needs. Almost all of them sell their agricultural products to traditional markets around the areas affected by the earthquake and liquefaction. 

(B) It can be reiterated that all the farmers affected by the earthquake and liquefaction did not complete their education in elementary school. Some even received primary school education but also dropped out or did not graduate, so their insights and knowledge were still very limited. Therefore, even though they are given guidance and counseling, it is difficult to accept it, especially with regard to things that are mystical or superstitious, especially those related to corpses in the ground due to the disaster.

---

## [Editor Report · Decision Letter 2]

5 Jan 2021

The Impact of Liquefaction Disaster on Farming Systems at Agriculture Land Based on Technical and Psychosocial Perspectives

PONE-D-20-00593R2

Dear Dr. Basir-Cyio,

We’re pleased to inform you that your manuscript has been judged scientifically suitable for publication and will be formally accepted for publication once it meets all outstanding technical requirements.

Kind regards,

Simon Clegg, PhD

Academic Editor

PLOS ONE

Additional Editor Comments:

Many thanks for resubmitting your manuscript to PLOS One

As you have addressed all the comments and the manuscript reads well, I have recommended it for publication

You should hear from the Editorial Office shortly.

It was a pleasure working with you and I wish you the best of luck for your future research

Hope you are keeping safe and well in these difficult times

Thanks

Simon

---

## [Editor Report · Acceptance letter]

8 Jan 2021

PONE-D-20-00593R2 

The Impact of Liquefaction Disaster on Farming Systems at Agriculture Land Based on Technical and Psychosocial Perspectives 

Dear Dr. Basir-Cyio:

I'm pleased to inform you that your manuscript has been deemed suitable for publication in PLOS ONE. Congratulations! Your manuscript is now with our production department. 

Kind regards, 

on behalf of

Dr. Simon Clegg 

Academic Editor

PLOS ONE